# Primary Functioning Hepatic Paraganglioma Treated by Laparoscopy: A Case Report

**DOI:** 10.3390/jcm11247282

**Published:** 2022-12-08

**Authors:** Chenhao Jiang, Chuwen Chen, Yongjie Zhou, Jian Yang, Jiayin Yang

**Affiliations:** 1Department of Liver Transplant Center, West China Hospital of Sichuan University, Sichuan University, Chengdu 610041, China; 2Laboratory of Liver Transplantation, Frontiers Science Center for Disease-Related Molecular Network, West China Hospital of Sichuan University, Sichuan University, Chengdu 610041, China; 3Department of Liver Surgery, West China Hospital of Sichuan University, Sichuan University, Chengdu 610041, China

**Keywords:** hepatic paraganglioma, hypertension in pregnancy, laparoscopy, case analysis

## Abstract

Paragangliomas are highly vascularised and often heritable tumors derived from the paraganglia. They are typically discovered in the retroperitoneal space as well as the head and neck region but are rarely encountered in the liver parenchyma. We report a case of a primary functioning hepatic paraganglioma and provide an up-to-date literature review of patients with such tumors. We present a case of functioning paraganglioma in a 34-year-old female patient who suffered a solitary lesion in her left lateral lobe with symptoms of hypertension since pregnancy. She did not have any family history and her pre-pregnancy examination was negative. An abdominal CT imaging revealed a 6.5 × 5.7 cm liver lesion in segments II and III. Laboratory investigations identified elevation in plasma-free catecholamines. With sufficient preoperative preparation, the patient underwent laparoscopic left hemihepatectomy. Immunohistochemical staining revealed Syn (+) tumor cell nests surrounded by S-100 sustentacular cells (+), providing a definitive diagnosis of paraganglioma. The patient recovered uneventfully without signs of recurrence during a 1-year follow-up period. Our case demonstrates that primary refractory hypertension in pregnancy should be screened for paraganglioma through abdominal ultrasound and plasma free catecholamines. On the other hand, laparoscopic surgery is technically safe and feasible for the treatment of patients with hepatic paragangliomas in favorable locations.

## 1. Introduction

Paragangliomas (PGLs), also known as extra-adrenal pheochromocytomas, are tumors derived from extra-adrenal chromaffin cells of the sympathetic paravertebral ganglia of the thorax, abdomen, and pelvis. Moreover, they also originate from parasympathetic ganglia located along the glossopharyngeal and vagal nerves in the neck and at the skull base [1]. As rare tumors, the clinical incidence of paragangliomas is approximately 3 per million per year [2]. 

These tumors are frequently associated with an excessive secretion of catecholamines, predisposing patients to multiple endocrine abnormalities of migraine-like headache, sweating, palpitations, episodic or sustained hypertension, pallor, or postural hypotension [3]. However, paragangliomas may be nonfunctional and synthesize no or only negligible catecholamines before producing space-occupying effects [4]. Notably, paragangliomas exhibit malignant behavior characterized by the presence of metastases at sites where chromaffin cells are normally absent, such as bones and lymph nodes [5]. Alpha-adrenergic receptor blockers before surgery are preferred as the first choice to minimize perioperative complications [6]. Furthermore, open resection for paragangliomas is routinely recommended, but laparoscopic resection can be performed for small, noninvasive paragangliomas in surgically favorable locations [7]. 

Here, we report a case of laparoscopic technique use in a 34-year-old woman who had hypertensive symptoms since pregnancy and was eventually diagnosed with primary functional hepatic paraganglioma. To our knowledge, there are no published cases of successful use of laparoscopy in a patient with hepatic functioning paraganglioma. Additionally, we performed a literature review on hepatic paraganglioma to update the clinical features of this rare disease. 

## 2. Case Presentation

A 34-year-old female patient presented to our hospital with a solitary lesion in her left lateral lobe (Figure 1A), who suffered symptoms of hypertension, palpitations, and dizziness since becoming pregnant. An ultrasound from the community hospital showed a 5.0 cm mass in the left lateral lobe. There was no serologic evidence of hepatitis B or C virus infection. Additionally, the blood tests showed excess catecholamines but normal alpha-fetoprotein (AFP) and vitamin K absence/antagonist-II (PIVKA-II) (Table 1), which were differentiated from typical hepatocellular carcinoma.

Abdominal enhanced computed tomography further revealed an irregularly shaped and poorly circumscribed lesion, measuring 6.5 × 5.7 cm in segments II and III of the liver with mild dilatation of intrahepatic bile duct. The single liver lesion was hypervascular and markedly enhanced in the early arterial phase, with slight washout in the portal phase (Figure 1B–D). The patient had no family history of related tumors. Her blood pressure measurements remained above 140/90 mmHg despite the use of beta-blockers, calcium-channel blockers, and diuretics in combination prior to admission. On admission, her blood pressure was 186/118 mmHg, and her heart rate was 84 beats per minute.

According to the anesthesiologist and endocrine consultations, an alpha-adrenergic receptor blocker (phenoxybenzamine, the starting dose was 10 mg b.i.d., the final dose was 20 mg b.i.d.) was used for 10 days preoperatively. In addition, a high-sodium diet and adequate rehydration were simultaneously performed to prevent unpredictable instability in blood pressure during surgery. After adequate preoperative preparation, a left hemihepatectomy was performed under laparoscopy depending mainly on the location of the tumor and the relatively mature laparoscopic technique. The lesion was completely encapsulated and had gross features of high vascularization from a serosal laparoscopic view (Figure 2A,B). In addition, the appearance of the adrenal glands was unremarkable on careful abdominal exploration. The Glisson intrathecal method was used, where the left hepatic artery and the left branch of the portal vein are dissected and separated within the Glisson sheath outside the hepatic parenchyma. Next, the left hepatic pedicle is closed with a bulldog, and the plane of hepatic parenchymal resection is determined according to the ischemic line. Then the liver parenchyma was dissociated along the hepatic incision line from superficial to deep with an ultrasonic scalpel, and the tangent surface was opened as much as possible to avoid deep debridement in a narrow space. After thinning the whole liver parenchyma, the left liver pedicle and left hepatic vein were then dissociated with a linear cutting closure. Ultimately, the specimen was removed from the suprapubic incision after being placed in a retrieval bag.

As the surgery progressed, it was noteworthy that a rapid decrease in blood pressure was present since the total dissection of the tumor. Postoperatively, the patient was transferred to the intensive care unit on a metaraminol infusion, and her blood pressure was stable at 113/70 mmHg. 

Grossly, the surgical specimen of paraganglioma displayed a solitary, fish-fleshy-like mass with a clear margin. Moreover, pathological examination of the surgical specimen indicated a solid tumor measuring 5.0 × 3.5 × 2.3 cm located in the left lateral lobe (Figure 2C,D). Microscopically, hematoxylin-eosin (H&E) staining showed that the tumor cells were arranged in small nests with the typical “zellballen pattern”, setting in a vascularly rich stroma. The immunohistochemical results were as follows: CgA (+), Syn (+), S-100 supporting cells (+), alpha-inhibin (−), CR (−), Hepa (−), CK8/18 (−), GPC−3 (+/−), PCK (−), HBME-1 (−), and A103 (−), which were consistent with a paraganglioma in postoperative pathology (Figure 3). The tumor was determined to be T2N0M0 according to the 8th edition of the American Joint Committee on Cancer (AJCC). The postoperative period was uneventful, and the oral feeding was resumed on postoperative day three with no complications. Within the initial 3-month follow-up, the blood pressure of the patient and the results of catecholamine tests in plasma returned to normal levels (Table 1). Serum normetanephrine and 3-methoxytyramine were 0.69 nmol/L and 2.81 pg/mL respectively, demonstrating notable decreases compared to the preoperative state. The patient showed no evidence of paraganglioma recurrence at the 1-year follow-up.

## 3. Discussion

The incidence of paraganglioma was 0.33 cases per 100,000 people per year. Moreover, pheochromocytoma and paraganglioma in pregnancy is rare, occurring in only 0.007% of all pregnancies [8]. Women generally have a higher incidence of paraganglioma than males in nearly every age group except children [9]. Ideally, PPGL should be diagnosed and treated prior to conception, especially in patients who have an associated germline pathogenic variant [10,11]. Paragangliomas may associate with refractory hypertension, with a prevalence of 0.2–0.6% in patients with hypertension [7], and present with clinical features of excess catecholamine production similar to intra-adrenal pheochromocytoma [12]. Other common symptoms of paragangliomas are headaches, sweating, hot flashes, palpitations, and pallor. In particular, the symptoms of catecholamine overdose are nonspecific and can overlap with other medical conditions including those of normal pregnancy, resulting in difficulties in diagnosis. 

Nevertheless, paragangliomas may be clinically silent and detected as incidentalomas. As a result, it is difficult to determine the correct diagnosis for hepatic paragangliomas, especially nonfunctioning paragangliomas. In the present case, refractory hypertension in a young woman without any family history since the beginning of pregnancy is noteworthy. On the other hand, preoperative biochemical testing showed elevated plasma norepinephrine, normetanephrine, and 3-methoxytyramine levels, which could be valuable in assisting the clinical diagnosis. Based on these clinical features, hepatic paraganglioma was suspected preoperatively. According to the patient’s negative pre-pregnancy examination, we simultaneously speculated that the tumor is stimulated by hormones, which allowed tumor growth and stimulated epinephrine production by tumor parenchymal cells during pregnancy [13,14]. Thus, refractory hypertension merits attention in liver space-occupying patients in clinical practice. Additionally, it is essential to differentiate from paraganglioma, phechromocytoma, and pregnancy-induced hypertension of pregnancy. Ultrasound presents an inexpensive, noninvasive, and convenient method that offers advantages for the initial characterization and localization of organic lesions. However, the final diagnosis still depends on the pathological results. 

To date, a total of 13 cases of hepatic paraganglioma have been reported by a thorough search of the PubMed database between 1995 and 2022 [15,16,17,18,19,20,21,22,23,24,25,26,27]. A retrospective chart review and summary of clinical features are presented in Table 2. There was no striking variation by gender, with slightly more women (8 cases). Subsequently, the most common clinical manifestations were asymptomatic (7 cases) and were often discovered occasionally during a physical examination. In addition, sweating (3 cases) and headaches (2 cases) are relatively common chief complaints. Only one patient had a recurrence located in the spleen and below the right posterior lobe of the liver three years after hepatic resection, with high noradrenaline level [22]. However, there are no literature reports respect to laparoscopic resection.

According to an improved understanding of catecholamine metabolism, measurements of plasma-free metanephrines or urinary fractionated metanephrines are commonly recommended [28,29]. Several case reports have shown some common imaging features in paragangliomas [18,22,24,25,30]. The masses are hyper-enhanced in the arterial phase and de-enhanced in the portal phase on contrast-enhanced CT scans. Additionally, this hypervascular lesion was hyperintense in T2-weighted MRI and hypointense in T1-weighted images. Advanced MRI imaging was utilized in paragangliomas. Diffusion-weighted imaging (DWI) was suggested to capture a gene mutation (SDH mutation) [31]. Additionally, perfusion MRI (DCE-MRI) was suggested to be useful for tumor differentiation [32]. In the present case, the tumor also displayed similar imaging characteristics. Moreover, 123I/131I-Metaiodobenzylguanidine (123I/131I-MIBG) and 68Ga-DOTATATE functional scanning are more sensitive and specific and are currently utilized for the detection of the origin of multiple tumors and metastases [33,34,35,36]. Collectively, PET/CT with 68Ga-DOTA-SSA has the highest diagnostic accuracy across imaging modalities. In addition, 123I-MIBG, 18F-FDOPA, and 18F-FDG could also play prominent roles, especially in specific clinical presentations or for evaluating therapeutic regimens [37]. 

Per the present clinical guidelines, perioperative medical management, including alpha-adrenergic receptor blockade, a high-sodium diet, and fluid intake, was used to prevent perioperative cardiovascular complications and severe hypotension after tumor removal [7]. Particularly in patients with heart or kidney failure risk, volume loading needs to be performed with caution. As mentioned in the literature, alpha-adrenergic receptor blockers should be started for at least seven days to restore normal blood pressure and reverse blood volume contraction preoperatively [38,39]. In addition, a target blood pressure of less than 130/80 mmHg with a heart rate of 60–70 bpm in a resting state seems justified, with adaptive adjustment for age and underlying diseases [7]. 

Surgery should always be the preferred treatment for locoregional hepatic paraganglioma. The combination of tumor location and maturation in laparoscopic technology was a pivotal factor enabling us to perform the left hemihepatectomy. Open resection for paragangliomas was recommended, but laparoscopic resection can be performed for small, noninvasive paragangliomas [12]. The laparoscopic approach is characterized by less pain, earlier recoveries, and shorter hospitalizations. Recently, advances in minimally invasive techniques and perioperative management have resulted in a higher proportion of patients eligible for laparoscopic liver resection [40]. Our case highlights that laparoscopic resection is feasible and safe for the treatment of paragangliomas with favorable locations and does not cause wild blood pressure fluctuation intraoperatively. Moreover, blood pressure, heart rate, and plasma glucose levels should be closely monitored for 24–48 h, aiding in the prevention of fluctuations in blood pressure and rebound hypoglycemia [41]. 

In clinical practice, the presence of these features in patients with paraganglioma and pheochromocytoma indicate a high probability of heredity, including a positive family history, multifocal, bilateral, and metastatic disease [42,43,44]. Of note, the highest frequencies of germline mutations are for SDHB (10.3%), SDHD (8.9%), VHL (7.3%), RET (6.3%), and NF1 (3.3%) [7]. Paragangliomas located in the sympathetic paraganglia are associated with lower overall survival rates and a higher rate of malignancy [45,46]. Approximately, 20–30% of the paragangliomas are malignant with poor survival [47,48], while the diagnosis of malignancy is difficult. It is noteworthy that although the surgical margins were histologically negative, the nature of paragangliomas deserves further attention. The accurate prediction of the biological behavior above these tumors remains difficult merely based on the pathological findings. Accordingly, malignancy in these tumors is defined by the presence of local invasion on gross and microscopic pathology or much more commonly by the presence of metastases, which may only be recognized several years after the completion of resection [49,50]. In this case, such uncertain biological potential tumors should be interpreted with caution, and a strict long-term follow-up is needed. 

## 4. Conclusions

In conclusion, routine screening of abdominal ultrasound and detection of catecholamines are necessary in atypical pregnancy-induced hypertension. Additionally, our review provides the first case in the world with a functioning hepatic paraganglioma to be resected laparoscopically. We contemplate that the safety and effectiveness of laparoscopic approaches could be thoroughly evaluated and discreetly selected to achieve complete resection. Appropriate expertise including anesthesia, endocrinology, and intensive care is ideally suited to manage patients perioperatively. Furthermore, we should be aware of the presence of hepatic paraganglioma even in asymptomatic individuals, and adequately master perioperative management for this situation. 

## Figures and Tables

**Figure 1 jcm-11-07282-f001:**
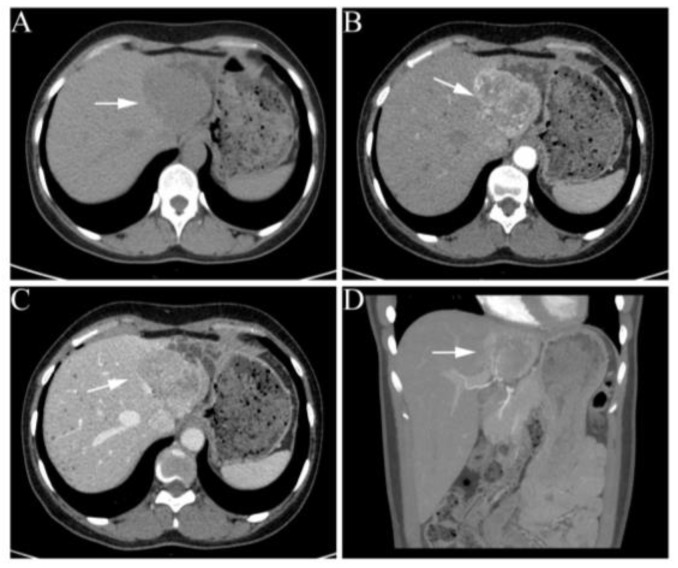
Abdominal contrast-enhanced CT images of the liver. (**A**) Non-enhanced scan shows a hypodense, well-marginated mass, measuring 6.5 × 5.7 cm. (**B**–**D**) A round inhomogeneous enhancement lesion in the early arterial phase and slight washout in the portal phase. White arrows indicate the liver lesion.

**Figure 2 jcm-11-07282-f002:**
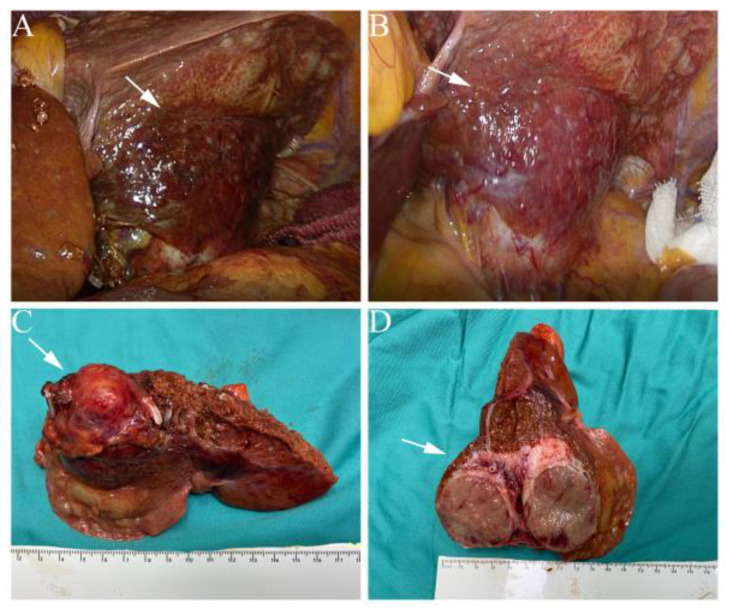
(**A**,**B**) Intraoperative images displayed a hypervascular mass in a subcapsular position. (**C**,**D**) Gross appearance and the cut surface of the mass. White arrows indicate the tumor.

**Figure 3 jcm-11-07282-f003:**
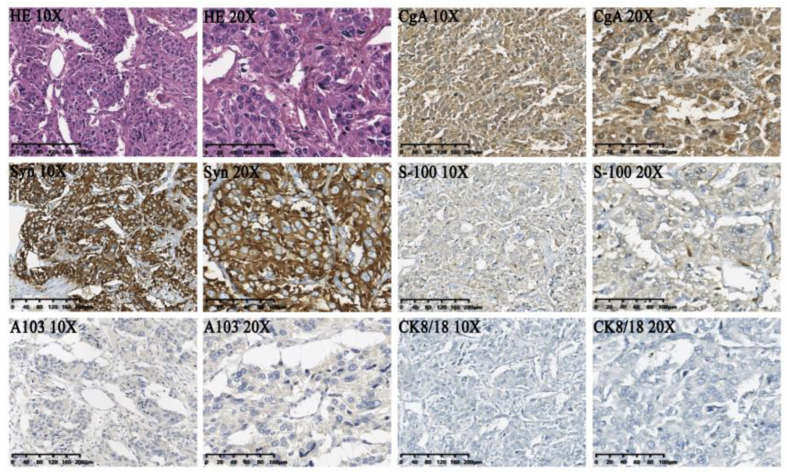
Histopathological features and immunohistochemical profiles of the paraganglioma: CgA (+), Syn (+), S-100 supporting cells (+), A103 (−), CK8/18 (−).

**Table 1 jcm-11-07282-t001:** Plasma catecholamines measurements confirmed the paraganglioma diagnosis.

	Preoperative Value	Status	Postoperative Value	Reference Value
Epinephrine	0.02		0.3	<0.34 nmol/L
Norepinephrine	38.61	↑	1.36	<5.17 nmol/L
Dopamine	0.16		0.05	<0.31 nmol/L
Metanephrine	0.07		0.13	<0.42 nmol/L
Normetanephrine	29.93	↑	0.69	<0.71 nmol/L
3-Methoxytyramine	26.59	↑	2.81	<18.40 pg/mL
AFP	3.83			<7.00 ng/mL
PIVKA-II	12			6.00–28.00 mAU/mL

↑: above the reference value range.

**Table 2 jcm-11-07282-t002:** Literature review of hepatic paragangliomas undergoing surgical resection.

First Author	Year	Country	Sex	Age	Size (cm)	Location	Chief Complaint	Operation Approach	Recurrence (Follow Up)
Jaeek [15]	1995	France	M	27	5.5 × 4 × 4	S VIII	Sweating	Open	No (1Y)
Reif [16]	1996	USA	F	42	5.0 × 4.5 × 3.8	S IV	Palpitations, sweating, headaches	Open	No (1.2Y)
Corti [17]	2002	Italy	M	46	8	S VII, VIII	Asymptomatic	Open	No (9Y)
Chang [18]	2006	China	M	37	6 × 6 × 4.5	S VI	Hypertension	Open	No (5Y)
Sharma [20]	2013	USA	F	80	11.7 × 10.5 × 8.8	S II, III, IV	Tachycardia, sweating, palpitations	Open	N/A
Hong [19]	2013	Korea	F	34	1.2 × 0.7	S VII	Asymptomatic	Open	No (1Y)
Xiao [21]	2015	China	M	59	6.1 × 5.8 × 5.5	S V, VII, VIII	Asymptomatic	Open	No (8Y)
You [22]	2015	China	F	47	3.6 × 3.4	S III	Asymptomatic	Open	Yes (4Y)
Liao [24]	2018	China	F	49	6.0 × 6.0	S VII, VIII	Asymptomatic	Open	No (2Y)
Kim [23]	2018	USA	M	16	7.4 × 4.5 × 3.0	S IV	Right eye pain with blurriness, headaches	Open	No (2.7Y)
Lin [25]	2019	China	F	41	4.5 × 3.5 × 4.0	S VII	Asymptomatic	Open	No (0.5Y)
Vella [27]	2021	Italy	F	29	12	S II, III	Asymptomatic	Open	No (1Y)
Li [26]	2022	China	F	47	3.8 × 3.2	Spieg lobe	Hyper menorrhagia, dizziness	Open	No (1Y)

Abbreviations: S-segment; Spieg-Spiegelman.

## Data Availability

The datasets used during the current study are available from the corresponding author on reasonable request.

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
