# Peer review of "Primary Functioning Hepatic Paraganglioma Treated by Laparoscopy: A Case Report"

_jcm, 2022, doi:10.3390/jcm11247282_

Round 1

Reviewer 1 Report

This is a case report of liver paraganglioma in a pregnant patient, successfully treated with laparoscopic surgery. This is an interesting case report because the case is rare and usually treated with open surgery as they suggested in the manuscript and a chart (table 2). This manuscript can be considered as a potential publication if they can address the suggestion. below.

Major comments

1. The role of imaging is scarcely mentioned. 

- As they suggested, the US should be done before the surgery for characterization and localization of the tumor. 

- Now Advanced MRI imaging has been now utilized. Diffusion-weighted imaging (DWI) has been suggested to capture gene mutation (SDH mutation) (Assessment of MR Imaging and CT in Differentiating Hereditary and Nonhereditary Paragangliomas. PMID: 33985956). Additionally, Perfusion MRI (DCE-MRI) has been suggested to be useful for tumor differentiation (Diagnostic Role of Diffusion-Weighted and Dynamic Contrast-Enhanced Perfusion MR Imaging in Paragangliomas and Schwannomas in the Head and Neck. PMID: 34446460).

Please briefly elaborate on these in the discussion or introduction.

2. There are gene mutations for hereditary paragangliomas (SDHA, SDHC, VHL, RET, NF1, TMEM127, and MAX genes). They suggested the heredity of this tumor in the abstract, but they did not mention it in the body of the manuscript. This should be done in the introduction or discussion.

3. The explanation of treatment options is scarce. In this case report, laparoscopic surgery was performed. Why did they choose laparoscopic surgery but not open surgery? The lesion seems large (5.0*3.5*2.3 cm). Is this due to the tumor size, location, or staging? 

And also, radiation therapy is one of alternative management options. Please mention how you select treatment options (laparoscopic, open surgery, or radiation therapy) depending on the various situations. 

Minor comments

Abstract

4. Please mention imaging findings (CT) when they made a diagnosis of liver paraganglioma.

Case presentation

5. US or MRI was performed in addition to CT? If done, please mention imaging findings.

6. Figure 1 B and C appears the same slice on the same CT phase. If they are so, please delete one of them. And also, if you have MRI or US imaging, please add these on figure 1.

Otherwise, This case is well written.

Discussion

7. Line 145-146, “Thus, refractory hypertension merits attention in liver space-occupation patients in clinical practice.” From this clinical scenario, isn’t Phechromocytoma considered in differential?

8. Line 163-164 “Several case reports have shown some common imaging features in paragangliomas.” This sentence needs a reference.

9. Line 165-167, “Additionally, this hypervascular lesion was demonstrated to be hyperintense in T2-weighted MRI and hypointense in T1-weighted images [18, 22, 24, 25, 30].” This imaging section needs to be more elaborated. As in comment in the major comments, Advanced MRI has been now used for tumor differential, gene mutation, and treatment assessment. Please mention this briefly with references.

10. Line 184-185, “Surgery should always be the preferred treatment for locoregional hepatic paraganglioma.” As mentioned in major comments, Based on what do you select surgery options? Is it based on tumor size, location, or staging? And also, radiation can be one of treatment options. Please mention briefly how you select the treatment options. 

Author Response

Response to Reviewer 1 Comments

Dear Editors and Reviewers,

Thank you for your letter and reviewers’ comments about our manuscript entitled “Primary Functioning Hepatic Paraganglioma Treated By Laparoscopy: A Case Report”. Those comments are all valuable and very helpful for revising and improving our paper, as well as the important guiding significance to our research. We have studied the comments carefully and have made corrections point-to-point which we hope meet with approval. Revised portions are highlighted in the paper. The main corrections in the paper and the responses to the reviewer’s comments are as follows:

Point 1: The role of imaging is scarcely mentioned.

- As they suggested, the US should be done before the surgery for characterization and localization of the tumor.

- Now Advanced MRI imaging has been now utilized. Diffusion-weighted imaging (DWI) has been suggested to capture gene mutation (SDH mutation) (Assessment of MR Imaging and CT in Differentiating Hereditary and Nonhereditary Paragangliomas. PMID: 33985956). Additionally, Perfusion MRI (DCE-MRI) has been suggested to be useful for tumor differentiation (Diagnostic Role of Diffusion-Weighted and Dynamic Contrast-Enhanced Perfusion MR Imaging in Paragangliomas and Schwannomas in the Head and Neck. PMID: 34446460).

Please briefly elaborate on these in the discussion or introduction.

Response 1: Thank you for your kind comments. The patient had undergone an ultrasound examination in the community hospital. Unfortunately, the patient could not provide an official report and the original image. She could only provide an oral description of the hepatic space-occupying lesion, which was a 5.0cm mass in the left liver. We have re-added the description of ultrasound in the case presentation and discussion (Line 59-60; Line 151-153).

As you said, advanced MRI imaging is necessary for gene mutation capture and tumor differentiation. We carefully read the relevant literature and learned a lot from it. In the background and discussion section, we added the relevant content and quoted the relevant literature in the discussion. (Line 174-177).  

Point 2: There are gene mutations for hereditary paragangliomas (SDHA, SDHC, VHL, RET, NF1, TMEM127, and MAX genes). They suggested the heredity of this tumor in the abstract, but they did not mention it in the body of the manuscript. This should be done in the introduction or discussion.

Response 2: Thank you for your kind comments. We have added a description of genetic variation for hereditary paragangliomas to the discussion section (Line 207-211).

Point 3: The explanation of treatment options is scarce. In this case report, laparoscopic surgery was performed. Why did they choose laparoscopic surgery but not open surgery? The lesion seems large (5.0*3.5*2.3 cm). Is this due to the tumor size, location, or staging?   

And also, radiation therapy is one of alternative management options. Please mention how you select treatment options (laparoscopic, open surgery, or radiation therapy) depending on the various situations.  

Response 3: Thank you for your kind comments. Our main consideration is the location of the tumor. The lesion in this patient was located in the left lateral lobe. As recommended by the guideline (The Southampton Consensus Guidelines for Laparoscopic Liver Surgery: From Indication to Implementation. PMID: 29064908), endoscopic excision of this site is mature. In addition, Our center carries out a large number of endoscopic operations every year, and the technology is relatively mature. Also, endoscopic surgery has the characteristics of less trauma and beautiful incisions. The patient requested a minimally invasive surgical resection after detailed informed consent. We added a description of this content (Line 86-87; Line 195-196). After the patient's outpatient follow-up to complete the PET CT, radiotherapy may be considered if there are metastatic lesions (As recommended by the “Personalized Management of Pheochromocytoma and Paraganglioma” PMID: 34147030).

Point 4: Please mention imaging findings (CT) when they made a diagnosis of liver paraganglioma.

Response 4: Thank you for your kind comments. We have added imaging findings (CT) to the abstract section. We added a description of this content (Line 20-21).

Point 5: US or MRI was performed in addition to CT? If done, please mention imaging findings.

Response 5: Thank you for your kind comments. The patient had undergone an ultrasound examination in the community hospital. Unfortunately, the patient could not provide an official report and the original image. She could only provide an oral description of the hepatic space-occupying lesion, which was a 5.0cm mass in the left liver. We have re-added the description of ultrasound in the case presentation (Line 59-60).

Point 6: Figure 1 B and C appears the same slice on the same CT phase. If they are so, please delete one of them. And also, if you have MRI or US imaging, please add these on figure 1.

Response 6: Thank you for your kind comments. We are very sorry for our negligence in uploading the portal phase Figure 1 C by mistake. We have made the correction according to your suggestion, and re-upload Figure 1.

Point 7: Line 145-146, “Thus, refractory hypertension merits attention in liver space-occupation patients in clinical practice.” From this clinical scenario, isn’t Phechromocytoma considered in differential?

Response 7: Thank you for your kind comments. As you said, it is important to include pheochromocytoma in the differential diagnosis, and we have revised it in this section according to your suggestion (Line 150-151).

Point 8: Line 163-164 “Several case reports have shown some common imaging features in paragangliomas.” This sentence needs a reference.

Response 8: Thank you for your kind comments. We have re-added references to this section (Line 171).

Point 9: Line 165-167, “Additionally, this hypervascular lesion was demonstrated to be hyperintense in T2-weighted MRI and hypointense in T1-weighted images [18, 22, 24, 25, 30].” This imaging section needs to be more elaborated. As in comment in the major comments, Advanced MRI has been now used for tumor differential, gene mutation, and treatment assessment. Please mention this briefly with references.

Response 9: As you said, advanced MRI imaging is necessary for gene mutation capture and tumor differentiation. We carefully read the relevant literature and learned a lot from it. In the background and discussion section, we added the relevant content and quoted the relevant literature in the discussion. (Line 174-177).   

Point 10: Line 184-185, “Surgery should always be the preferred treatment for locoregional hepatic paraganglioma.” As mentioned in major comments, Based on what do you select surgery options? Is it based on tumor size, location, or staging? And also, radiation can be one of treatment options. Please mention briefly how you select the treatment options.

Response 10: Thank you for your kind comments. Our main consideration is the location of the tumor. The lesion in this patient was located in the left lateral lobe. As recommended by the guideline (The Southampton Consensus Guidelines for Laparoscopic Liver Surgery: From Indication to Implementation. PMID: 29064908), endoscopic excision of this site is mature. In addition, Our center carries out a large number of endoscopic operations every year, and the technology is relatively mature. Also, endoscopic surgery has the characteristics of less trauma and beautiful incisions. The patient requested a minimally invasive surgical resection after detailed informed consent. We added a description of this content (Line 195-196).

After the patient's outpatient follow-up to complete the PET CT, radiotherapy may be considered if there are metastatic lesions (As recommended by the “Personalized Management of Pheochromocytoma and Paraganglioma” PMID: 34147030).

Thank you very much for your comments and suggestions!     

Sincerely yours,

Jiayin Yang, M.D. & Ph.D.

Reviewer 2 Report

This is an interesting report of laparoscopic resection of hepatic paraganglioma. Well written. 

Comments - 

1. Did your patient undergo systemic evaluation by a functional imaging (PET CT or FDOPA)? Was it at least done postoperatively? Please comment. 

2. All patients with paragangliomas should undergo genetic evaluation for SDHB and other mutations. What was the genetic status of your patient? 

Author Response

Response to Reviewer 2 Comments

Dear Editors and Reviewers,

Thank you for your letter and reviewers’ comments about our manuscript entitled “Primary Functioning Hepatic Paraganglioma Treated By Laparoscopy: A Case Report”. Those comments are all valuable and very helpful for revising and improving our paper, as well as the important guiding significance to our research. We have studied the comments carefully and have made corrections point-to-point which we hope meet with approval. Revised portions are highlighted in the paper. The main corrections in the paper and the responses to the reviewer’s comments are as follows:

Point 1: Did your patient undergo systemic evaluation by a functional imaging (PET CT or FDOPA)? Was it at least done postoperatively? Please comment.  

Response 1: Thank you for your kind comments. As you said, systemic evaluation by functional imaging was necessary. After the operation, we routinely suggested PET CT to the patient in order to evaluate the lesions elsewhere in the body. The patient complained that she was still lactating and did not want to do another CT scan, so she refused the suggestion of PET CT. However, routine abdominal ultrasound and biochemical testings were carried out as usual. We will continue the functional imaging recommendations during our recent outpatient review.

Point 2: All patients with paragangliomas should undergo genetic evaluation for SDHB and other mutations. What was the genetic status of your patient?

Response 2: Thank you for your kind comments. As you said, genetic evaluation was necessary for patients with paragangliomas. The detection of gene mutation has been routinely informed to the patient, but genetic testing is not covered by medical insurance, and the testing price is relatively high. Unfortunately, the patient refused the genetic evaluation due to the financial costs and only agreed to a routine review.

Thank you very much for your comments and suggestions!                                                                                                                                                                                                                                       Sincerely yours,

Jiayin Yang, M.D. & Ph.D.